

# Introgression between ecologically distinct species following increased salinity in the Colorado Delta-Worldwide implications for impacted estuary diversity

Clive L.F. Lau and  David K. Jacobs

Department of Ecology and Evolutionary Biology, University of California, Los Angeles, CA, United States of America

Corresponding author
David K. Jacobs, djacobs@ucla.edu

## ABSTRACT

We investigate hybridization and introgression between ecologically distinct sister species of silverside fish in the Gulf of California through combined analysis of morphological, sequence, and genotypic data. Water diversions in the past century turned the Colorado River Delta from a normal estuary to a hypersaline inverse estuary, raising concerns for the local fauna, much of which is endangered. Salinity differences are known to generate ecological species pairs and we anticipated that loss of the fresher-water historic salinity regime could alter the adaptive factors maintaining distinction between the broadly distributed Gulf-endemic *Colpichthys regis* and the narrowly restricted Delta-endemic *Colpichthys hubbsi*, the species that experienced dramatic environmental change. In this altered environmental context, these long-isolated species (as revealed by Cytochrome *b* sequences) show genotypic (RAG1, microsatellites) evidence of active hybridization where the species ranges abut, as well as directional introgression from *C. regis* into the range center of *C. hubbsi*. Bayesian group assignment (STRUCTURE) on six microsatellite loci and multivariate analyses (DAPC) on both microsatellites and phenotypic data further support substantial recent admixture between the sister species. Although we find no evidence for recent population decline in *C. hubbsi* based on mitochondrial sequence, introgression may be placing an ancient ecological species at risk of extinction. Such introgressive extinction risk should also pertain to other ecological species historically sustained by the now changing Delta environment. More broadly, salinity gradient associated ecological speciation is evident in silverside species pairs in many estuarine systems around the world. Ecological species pairs among other taxa in such systems are likely poorly understood or cryptic. As water extraction accelerates in river systems worldwide, salinity gradients will necessarily be altered, impacting many more estuary and delta systems. Such alteration of habitats will place biodiversity at risk not only from direct effects of habitat destruction, but also from the potential for the breakdown of ecological species. Thus, evolutionary response to the anthropogenic alteration of salinity gradients in estuaries merits investigation as the number of impacted systems increases around the globe, permitting parallel study of multiple systems, while also permitting a conservation management response to help preserve this little championed component of biodiversity.

## INTRODUCTION

Damming and water extraction in river systems are accelerating around the globe (*Nilsson et al., 2005*; *Liermann et al., 2012*), impacting estuarine and deltaic settings downstream (*Bonetto, Wais & Castello, 1989*; *Lu & Siew, 2006*; *Yang et al., 2005*). Physical changes include loss of freshwater, elimination of peak flood flows, loss of sediment supply, and alteration of the nature and position of salinity gradients (*Nilsson & Berggren, 2000*; *Carriquiry & Sánchez, 1999*). Such changes necessarily influence the adaptive context under which local species evolve, potentially reducing the selective pressures maintaining species differences and placing ecological species at risk of elimination through introgression. In order to better understand the risks such changes pose, we examine recent introgression in a Pliocene-age species pair of silverside fishes of the genus *Colpichthys* in the northern Gulf of California (Gulf hereafter) where the Colorado River enters the Ocean. Following 20th century dam constructions, virtually no river flow reaches the Colorado River Delta (*Glenn et al., 1996*). As a consequence of this dramatically modified state, this system may serve as a harbinger of evolutionary impacts in the estuaries of the world's major river systems.

*Colpichthys*, a little studied genus of the silverside family Atherinopsidae, inhabits near-shore settings of the northern part of the Gulf and consists of two endemic species, *Colpichthys regis* (*Jenkins & Evermann, 1889*) and *Colpichthys hubbsi* (*Crabtree, 1989*). These are listed on the IUCN Red List as Near Threatened and Endangered, respectively (*Findley, Collette & Espinosa, 2010a*; *Findley, Collette & Espinosa, 2010b*). The Colorado River enters the Gulf at its northernmost extremity where it forms an extensive delta and estuary system. The Colorado Delta (Delta hereafter) habitat of *C. hubbsi* was historically distinct from the other marine estuaries of the semiarid Gulf where we recovered *C. regis* samples (*Crabtree, 1989*; *Hastings & Findley, 2007*). Tidal channels in the Delta experience a much greater tidal amplitude of up to 10 m, and silt as opposed to clay content is much higher in the Delta relative to other estuaries. In the 19th century, the Delta received continuous freshwater flow and extensive flood water during the spring and summer melting of mountain snows in the headwaters. These flows were eliminated through 20th century damming leading to a shift from a typical brackish to a hypersaline inverse estuary (*Lavín & Sánchez, 1999*). The difference in salinity, sediment supply and food chain necessarily impose significantly different adaptive regimes on taxa in the Delta relative to other Gulf estuaries (*Swift et al., 2011*); as a result of the shifting environment, these differences may disappear.

Through the study of introgressive process in *Colpichthys* we hope to inform issues of evolution and endangerment of the northern Gulf fauna associated with the potential for introgression in ecological species in the face of habitat change. Direct and climate driven anthropogenic impacts increasingly threaten estuarine faunas globally across a broader suite of river systems. In the remainder of this introduction we consider: similar contexts
that have produced ecological species, the potential extirpation of species via introgression, the relevant detail regarding the study taxon *Colpichthys* as well as the suite of data and methods employed in phylogeny reconstruction, and assessment of historic demography and introgression.

Marine taxa that invade fresh water experience strong selection (*Lee, 2016*). In these cases, advantages in food availability or predator avoidance can overcome the selective costs and rapid evolution of tolerance to fresh water often ensues. Ecological divergence between fresh and marine water commonly occurs in the lower reaches of rivers, and is especially well established in silversides. Ecological diversification and speciation by salinity differences occur in geographically isolated and taxonomically separate silverside genera from Brazil, Australia, the Mediterranean, and the Eastern US (*Bamber & Henderson, 1988*; *Beheregaray & Sunnucks, 2001*; *Fluker, Pezold & Minton, 2011*; *Johnson, Watts & Black, 1994*; *Klossa-Kilia et al., 2007*; *Francisco et al., 2006*; *Olsen, Anderson & McDonald, 2016*; *Trabelsi et al., 2002*). The strong selection pressures between fresh and salt water, the frequent ecological diversification in silversides, and the historic difference in habitats between the Delta and the Gulf as noted above all support an ecological species interpretation of the *C. regis*/*C. hubbsi* split. As the loss of fresh water input to the Delta disrupts the local salinity gradient, breakdown of the ecological factors supporting distinction of these species seems likely.

Introgressive hybridization involves the transfer of alleles from one species to the gene pool of another species through repeated backcrossing of hybrids (*Heiser, 1973*; *Harrison & Larson, 2014*). This process can result from the dissolution of selective regimes that maintain ecological species and has a number of evolutionary consequences. Introgression could allow for the spread of adaptive or deleterious alleles from one species to the other across a hybrid zone. Alternatively, it may promote the homogenization of the two species, and in extreme cases, could lead to the extinction of one or both parental types through effects such as demographic or genetic swamping (*Todesco et al., 2016*). The hydrologic changes in the Delta make *C. hubbsi* a likely candidate for introgressive hybridization with *C. regis*. *C. regis* has a range rendered discontinuous by the Colorado River Delta at the northern end of the Gulf where *C. hubbsi* is locally endemic. *C. regis* and *C. hubbsi* co-occur only at the southwestern border of the Delta where it meets the Baja coast (Fig. 1A). In this study, we test whether hybridization occurs in this region of overlap and whether introgression is occurring between the species.

In the following, we compare the external and internal morphology of specimens recovered from *C. regis* and *C. hubbsi* populations to characterize the variation across both species to help establish baseline for discovering potential hybrids. We also investigate molecular evidence from mitochondrial and nuclear sequences as well as microsatellite genotypes to detect genetic introgression across the species barrier. In addition to investigating the hybridization, with the available genetic data we performed a number of population genetics analyses to infer genetic diversity and demographic histories of the two species.
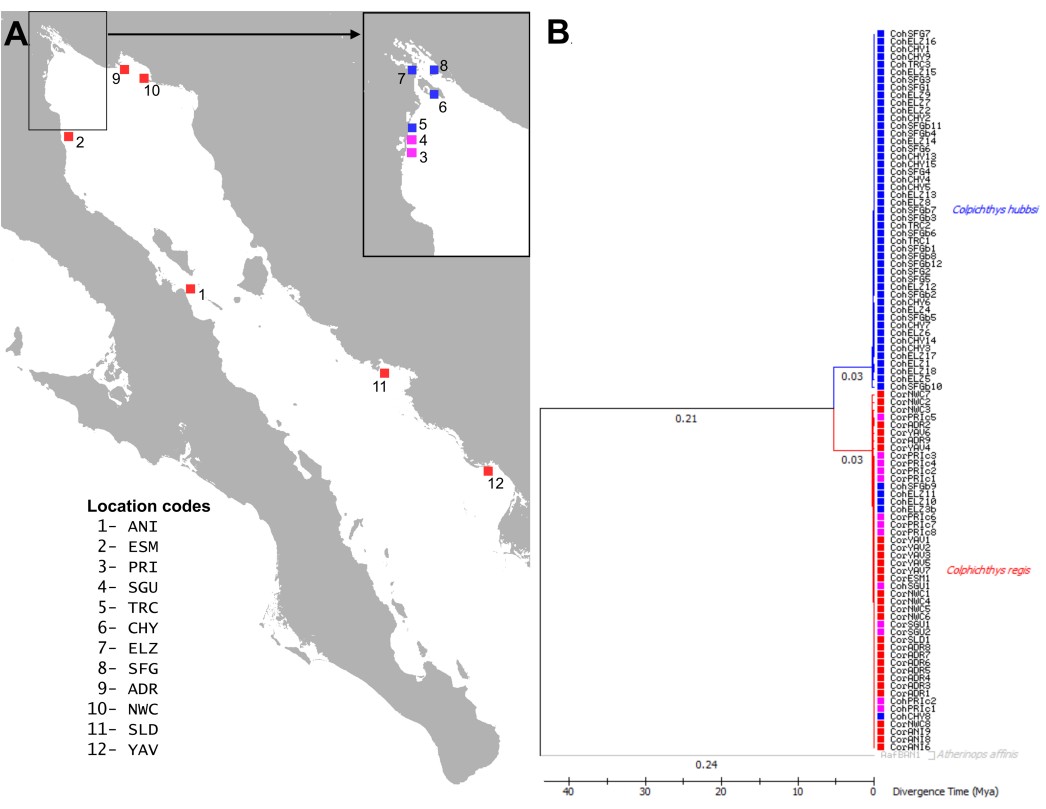

**Figure 1** **Map and Phylogeography of *Colpichthys*.** (A) Map of the coastline of the Gulf of California with sampling localities, color coded by species distribution. *C. hubbsi* native populations are colored blue; *C. regis* native populations are colored red; and populations where both species co-occur are colored magenta. Location codes are summarized in Table 1 and used in text throughout. (B) Time calibrated Maximum-Likelihood (ML) tree of cytochrome *b* (872 bp). Branch lengths (substitutions per site) are indicated below the branches. One specimen of *Atherinops affinis* was used as outgroup to root the tree. Branch labels show branch lengths (substitutions/site).

## MATERIALS AND METHODS

### Sample collection

Specimens of the genus *Colpichthys* were collected by seine from estuarine habitats along the coast of the Northern Gulf of California and within the Colorado River Delta in the years 2005, 2006, 2007, and 2011 (Table 1). Collections were carried under Mexican Federal Collecting Permit (Permiso de Pesca de Fomento) DGOPA 14253. 101005.6950, and its extension DGOPA 06435.210606.2640, issued to Findley and Jacobs by the Comisión Nacional de Acuacultura y Pesca of the Secretaría de Agricultura, Ganadería, Desarrollo Rural, Pesca y Alimentación (SAGARPA). Collected specimens were preserved in 95% ethanol in the field and stored at −20 °C upon returning to the laboratory for use in both molecular and morphological analyses. For this study, we divided the coastline of the Northern Gulf of California into 4 regions: (1) the Delta, native to populations of *C. hubbsi*; (2) the Baja Coast and (3) the Sonora Coast, native to populations of *C. regis*; and (4) the Delta Edge, the putative hybrid zone. A full list of collection sites along with the abbreviated

**Table 1  Collection localities for the modern specimens.** List of collection localities for the modern specimens.

| Locality | Code | Region | Sampling date | Coordinates | N = |
|---|---|---|---|---|---|
| Las Animas | ANI | Baja Coast | 4 May 2007 | 28°47.855N–113°20.894W | 9 |
| Estero Santa María | ESM | Baja Coast | 10 Nov 2005 | 30°44.73′N–114°42.01W | 1 |
| Estero Primero | PRI | Delta Edge | 22 Sep 2006 | 31°11.903N–114°53.437W | 10 |
| Estero Segundo | SGU | Delta Edge | 24 Sep 2006 | 31°15.355N–114°53.011W | 3 |
| Estero Tercero | TRC | Colorado River Delta | 16 Jun 2011 | 31°17.354N–114°54.831W | 3 |
| Estero Chayo | CHY | Colorado River Delta | 3 Dec 2005 | 31°40.119N–114°41.529W | 15 |
| "Port Elizabeth" | ELZ | Colorado River Delta | 3 Dec 2005 | 31°49.405N–114°49.566W | 18 |
| "Shrimp Farm" El Golfo | SFG/SFGb | Colorado River Delta | 2 Dec 2005/15 Jun 2011 | 31°46.480N–114°34.931W | 7 12 |
| Bahía Adair | ADR | Sonora Coast | 14 Jun 2011 | 31°32.244N–113°58.910W | 9 |
| Northwest of Cholla | NWC | Sonora Coast | 14 Jun 2011 | 31°27.822N–113°37.898W | 8 |
| Estero del Soldado | SLD | Sonora Coast | 6 Oct 2006 | 27°57′26″N–110°58′48″W | 1 |
| Yavaros | YAV | Sonora Coast | 5 Mar 2006 | 26°40′42″N–109°29′36″W | 7 |
| | | | | | Total = 103 |

**Table 2  List of museum specimens.** List of "chistorical" specimens from the Natural History Museum of Los Angeles County collection (LACM) and Scripps Institution of Oceanography collection (SIO).

| Collection ID | Collection date | Locality | Locality code | Coordinates | Species | Feature examined | N = |
|---|---|---|---|---|---|---|---|
| SIO 81-158 | 19 Jan 1973 | East of Isla Montague | EIM | 31°46.0′N–114°42.5′W | *C. hubbsi* | External morphology | 2 |
| SIO 81-161 | 15 Apr 1973 | North of Isla Montague | NIM | 31°49.0′N–114°48.5′W | *C. hubbsi* | External morphology | 1 |
| SIO 81-156 | 17 Dec 1972 | West of Isla Montague | WIM | 31°44.0′N–114°48.0′W | *C. hubbsi* | External morphology | 1 |
| SIO 68-46 | 20 Jan 1968 | North of San Felipe | SFE | 31°0.0′N–114°52.0′W | *C. hubbsi* [a] | External morphology | 16 |
| SIO 60-484 | 5 Dec 1960 | West from Empalme | EMP | 27°55.0′N–110°55.0′W | *C. regis* | External morphology | 3 |
| SIO 63-532 | 17 Aug 1963 | Kino Bay | KIN | 28°51.5′N–112°1.5′W | *C. regis* | External morphology | 1 |
| SIO 63-531 | 17 Aug 1963 | Laguna La Cruz | KNV | 28°48.0′N–111°54.5′W | *C. regis* | External morphology | 1 |
| LACM 35730 | 23 May 1974 | Guaymas | GYM | 27°55′N–110°57′W | *C. regis* | Vertebral morphology | 3 |

Notes.
[a]Specimens were classified as *C. hubbsi* on label, but all except one specimen from this locality display the external morphology of *C. regis*. (For details, see Table S1.)

designation for each site are shown in Table 1. In addition to the field-collected specimens, formalin-fixed specimens from the Natural History Museum of Los Angeles County (LACM) and Scripps Institution of Oceanography were examined for their morphology to confirm species identification. A list of these "historical" specimens is included in the supplementary materials (Table 2).

## Phenotypic analyses

The dorsal-lateral scale counts, vertebral counts, hemal spine morphology, and the position of the first dorsal fin were used as species-distinguishing features for this study; these features were chosen based on the species description of *Colpichthys hubbsi* (*Crabtree, 1989*). The first dorsal fin position is assessed based on the linear length from the snout to the first dorsal fin origin, normalized by the standard length. Length measurements were taken with dial calipers. Morphometric and meristic counts were performed according to *Hubbs, Lagler & Smith (2004)*. Radiographs of specimens were taken for examination of their

vertebral counts and hemal spine morphology. Six specimens from location NWC and three specimens from location YAV were too small to accurately assess their hemal spine morphology; they were thus excluded from the hemal spine data set.

## Molecular procedures

Genomic DNA was isolated from caudal peduncle muscle tissues using the DNeasy Blood and Tissue Kit (Qiagen, Inc., Valencia, CA, USA). The mitochondrial cytochrome *b* gene was amplified by PCR using primers AJG15 and H5 (*Akihito et al., 2000*) under the following cycling conditions: 95 °C for 2 min; 40 cycles of 95 °C for 30 s, 51 °C for 30 s, followed by 72 °C for 90 s; then finally 72 °C for 10 min. PCR products were size-checked on 1.5% agarose gel and cleaned with ExoSap before Sanger sequencing with BigDye Terminator v3.1. Samples were then submitted to DNA Analysis Facility on Science Hill at Yale University for capillary electrophoresis.

Amplification of the nuclear RAG1 gene was carried out using primers RAG1-2533F and RAG1-4090R (*López, Chen & Ortí, 2004*) under the following cycling conditions: 95 °C for 2 min; 40 cycles of 95 °C for 1 min, 51 °C for 1 min with an incremental increase of 0.5 °C per cycle until 55 °C , followed by 72 °C for 90 sec; then finally 72 °C for 10 min. Post-PCR processing and sequencing of the RAG1 gene were performed as described above. From the sequences, we were able to identify four polymorphic sites; several individuals display ambiguity in base calling at such sites, and these were tagged as putative heterozygotes. A total of four RAG1 variants were inferred from the sequence data. Homozygous individuals were used to confirm the existence of three RAG1 variants; the remaining RAG1 variant was inferred from a heterozygous individual. To confirm the gametic phase of these RAG1 sequences, we employed allele-specific polymerase chain reaction (ASPCR) as detailed in *Wu et al. (1989)*. Three allele-specific primers (T717F: 5′-CTACAAAATCTTCCAGGAT-3′, T770R: 5′-TTTATCTAAGGCTGCCCTCCAGA-3′, and A909F: 5′-AACTGGTGCCCTCAGAAGAA-3′) were designed to uniquely amplify one of the two diploid copies of RAG1 in the putative heterozygotes. The 3′ terminating nucleotide of these primers are designed to specifically anneal at the predetermined polymorphic site, thus enabling unique amplification of alleles and determination of the RAG1 genotypes of the heterozygotes.

Six microsatellite loci (Odont08, Odont09, Odont11, B18, B19, B39) were amplified for genotyping. Primer sequences for loci Odont08, Odont09, and Odont11 were obtained from *Beheregaray & Sunnucks (2000)*. Primer sequences for loci B18, B19, B39 were obtained from *Byrne & Avise (2009)*. Loci which can be easily differentiated by lengths were multiplexed two at a time using QIAGEN Multiplex PCR kits with 6-FAM labeled M13 primer (*Boutin-Ganache et al., 2001*). Only the forward primers of each locus are 5′ tagged with the M13 sequence, and reactions were performed under the following cycling conditions: 95 °C for 15 min; 25 cycles of 94 °C for 30 s, 55 °C for 90 s, 72 °C for 1 min; then another 25 cycles of 94 °C for 30 s, 50 °C for 90 s, 72 °C for 1 min; and finally 60 °C for 30 min. Diluted PCR products were submitted to the UCLA GenoSeq core facility for genotyping on an ABI3730 (Applied Biosystems, Foster City, CA, USA).

## Molecular analyses

All sequences were aligned, trimmed, and analyzed using Geneious Pro ver. 5.5.6 (*Drummond et al., 2011*). A Maximum Likelihood tree of Cyt *b* sequence was made with MEGA7 (*Kumar, Stecher & Tamura, 2016*) and rooted using *Atherinops affinis* as the outgroup. A time-calibrated tree was inferred using the Reltime method (*Tamura et al., 2012*) and the Tamura-Nei model (*Tamura & Nei, 1993*). The timetree was computed using one calibration constraint at the node of the split between *C. regis* and *C. hubbsi*; the calibration point of 4.83Ma was obtained from the age of the ash bed within the Bouse Formation, the rock unit from which two fossilized *C. regis* were documented (*Todd, 1976*; *Spencer et al., 2013*).

Microsatellite data were obtained from 101 out of 103 specimens and imported into Geneious for allele scoring and analyzed using STRUCTURE ver. 2.3.4 (*Pritchard, Stephens & Donnelly, 2000*). STRUCTURE is a program that uses a Bayesian clustering approach to infer population structure from genotypic data. Under an admixture model of K populations or genetic groupings, STRUCTURE computes the $q$-value, a quantity between 0 and 1, that reflects the proportions of an individual's genome originating from a certain population. Hybridization can then be inferred based on the computed $q$-value; for example, first-generation (F1) hybrids of two populations ($K = 2$) are expected to have a $q$-value close to 0.5 (*Vähä & Primmer, 2006*). We ran STRUCTURE analyses using an admixture model with 1,000,000 MCMC generations after a burn-in period of 100,000. Sampling locations were incorporated as prior parameters in the *LOCPRIOR* model to improve clustering for our data set; the *LOCPRIOR* model is preferred when there are weak signals of structure due to low number of markers analyzed (*Hubisz et al., 2009*). Both correlated and independent allele frequency models were used with number of groups set from $K = 1$ to $K = 4$; for each $K$ value, analysis was repeated ten times. Of the two allele frequency models, the correlated model provides greater power in differentiating closely related populations (*Falush, Stephens & Pritchard, 2003*). The optimal $K$ value was informed by $\Delta K$ (*Evanno, Regnaut & Goudet, 2005*) as computed using Structure Harvester (*Earl & VonHolt, 2012*). Output of the STRUCTURE analyses were summarized in CLUMPP ver. 1.1.2 (*Jakobsson & Rosenberg, 2007*) and visualized in DISTRUCT ver. 1.1 (*Rosenberg, 2004*).

## Multivariate analysis

We further analyzed our phenotypic and microsatellite data by discriminant analysis of principal components (DAPC) using the R-package *adegenet* (*Jombart & Ahmed, 2011*). This analysis is used to help describe the phenotypic and genetic clustering of our specimens. Cluster priors for DAPC were identified by *k*-means clustering; based on BIC scores we selected the 2-cluster model to use for the DAPC. To perform DAPC, the program transforms the data into principal components, which is then analyzed through discriminant analysis. Repeated cross-validation was used to evaluate the optimal number of principal components to retain for the discriminant analyses to avoid overfitting of data. For the phenotypic analysis, we used measurements of the four species-distinguishing features as our data and retained one PC (91.28% of the cumulative variance) for generating the morphology discriminant function. For the microsatellite analysis, we used the allele

frequencies of the six loci as our data and retained 20 PCs (66.35% of the cumulative variance) for generating the microsatellite discriminant function.

## Demographic histories

To explore the demographic histories of *Colpichthys*, Cyt *b* data sets were separated by species according to the Cyt *b* gene tree (Fig. 1B). Forty-four sequences from *C. hubbsi* and 52 sequences from *C. regis* were analyzed for their haplotype diversity. Median-joining haplotype networks were drawn in POPART ver. 1.7 (*Leigh & Bryant, 2015*). Cyt *b* haplotypes were examined by various neutrality tests—Tajima's *D* (*Tajima, 1989*), Fu and Li's *D\** and *F\** (*Fu & Li, 1993*), Fu's *Fs* (*Fu, 1997*), and $R_2$ (*Ramos-onsins & Rozas, 2006*)—in DnaSP v. 5.10.01 (*Librado & Rozas, 2009*). *P*-values for these test statistics were obtained by coalescent simulations of 1,000 replicates, conditional on the number of segregating sites. Pairwise differences were analyzed by mismatch distributions in Arlequin ver. 3.1 (*Excoffier, Laval & Schneider, 2005*); coalescent simulations were run with 10,000 bootstrap replicates. Past demographic expansions can be inferred from haplotypes based on the mismatch distribution of the pairwise differences; an uneven, multi-modal distribution is expected for populations under demographic equilibrium, whereas a "smooth" uni-modal distribution is expected for populations that had undergone demographic expansion (*Rogers & Harpending, 1992*). The observed mismatch distributions of the two species were compared to the expected mismatch distribution under a sudden expansion model using the sum of square deviation (SSD) and the raggedness index.

# RESULTS

## Pure-bred Specimens of *C. regis* and *C. hubbsi* are morphologically distinct

Consistent with the findings of *Crabtree (1989)*, our morphological analysis based on both modern and historical specimens, summarized in Fig. 2, shows that *C. regis* (green and gold) differs from *C. hubbsi* (pink) in having higher dorsal-lateral scale counts, higher vertebral counts, and shorter snout to first dorsal fin origin distance. In addition, the symphases of the hemal spines of the anterior-most caudal vertebrae of *C. regis* are modified into an expanded process (Fig. S1), forming a funnel-like structure antero-posteriorly. The number of modified hemal spines within *C. regis* outside of the Delta Edge varies between 5 to 11; the hemal spines of *C. hubbsi* in the Delta appears to show no signs of such modification (Fig. 2D). Outside the putative hybrid zone at the Delta Edge, individuals are relatively easily classified to species based on morphological differences. However, more continuous variation is evident within the Delta Edge region.

## Introgression of Cytochrome b and RAG1 haplotypes

Maximum-Likelihood phylogeny reconstruction of Cyt *b* sequence reveals ∼6% divergent clades corresponding to the two species of *Colpichthys* (Fig. 1B). *C. hubbsi* haplotypes were recovered from the Delta region, while Baja and Sonoran Coast samples contained only *C. regis* haplotypes. Both species were expected to cohabit the localities at the southwestern border of the Delta (PRI, SGU, and TRC), but none of these locations contain a mixture of

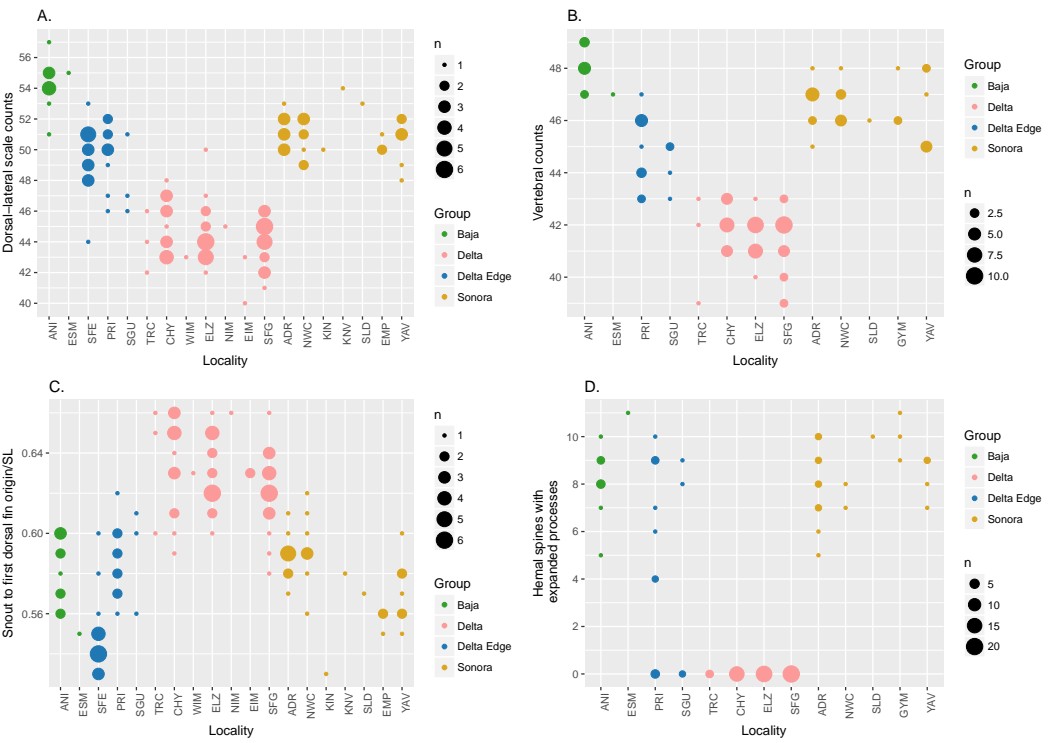

**Figure 2 Dot plots of morphological data.** Dot plots showing external (A and C) and vertebral (B and D) morphological data; size of dots is proportional to the number of specimens (n). "Delta Edge" is the hypothesized hybrid zone; "Delta" is the territory for *C. hubbsi*; "Baja" and "Sonora" are the territories for *C. regis*. Refer to Tables 1 and 2 for location codes of modern and historical specimens, respectively. Six specimens from NWC and three specimens from YAV were excluded from the hemal spine data set (D) as these specimens were too small to reliably identify the presence/absence of the expanded process on the hemal spines (See Fig. S1).

both haplotypes. PRI and SGU fish are all *C. regis* haplotypes, while the more Delta internal TRC sample is exclusively *C. hubbsi* haplotypes, albeit with limited sample sizes. Within the Delta, five specimens collected from predominantly *C. hubbsi* populations—ELZ, CHY, and SFG—were found to possess the *C. regis* mitochondrial haplotype; these specimens are morphologically indistinguishable from other *C. hubbsi* specimens.

We identified four RAG1 variants (labeled as $H_1$, $H_2$, $H_3$, and $R$; Table 3) based on four polymorphic sites in the RAG1 sequence data (975 bp). We find that $H_1$, $H_2$, and $H_3$ are common in populations of *C. hubbsi* in the Delta, while $R$ is fixed in all populations of *C. regis* in the Baja Coast and Sonora Coast regions. Thus, the two species are well differentiated by their RAG1 genotypes, with the diagnostic SNPs at positions 717 and 909. Six specimens, heterozygous at both of these positions, are flagged as putative hybrids. Four of the six hybrids occur at the Delta Edge, while the remaining two are found within the Delta proper (Table 4). No RAG1 heterozygotes are detected in any of the Baja and Sonoran collection sites.

As noted above, morphology of the anterior-most caudal hemal spines provides a phenotypic difference between the two species of *Colpichthys*. The non-modified hemal

**Table 3** **RAG1 haplotypes.** Variants of RAG 1. $H_1$, $H_2$, $H_3$ are common in populations of *C. hubbsi* in the Delta; $R$ is fixed in all populations of *C. regis*. SNPs at positions 717 and 909 (bolded) are species-specific and heterozygotes at these positions are used to identify putative hybrids.

| RAG1 Variants | Position 495 | Position 717 | Position 770 | Position 909 |
|---|---|---|---|---|
| $H_1$ | G | **C** | G | **G** |
| $H_2$ | G | **C** | T | **G** |
| $H_3$ | A | **C** | T | **G** |
| $R$ | G | **T** | G | **A** |

**Table 4** **Distribution of RAG1 heterozygotes.** Counts of RAG1 heterozygotes and non-heterozygotes at positions 717 and 909 at each of the Delta and Delta Edge collection sites.

| | | Heterozygotes | Non-heterozygotes | Het./non-het. ratio |
|---|---|---|---|---|
| Delta | Port Elizabeth (ELZ) | 0 | 18 | 0 |
| | Shrimp Farm (SFG) | 1 | 15 | 0.067 |
| | Estero Chayo (CHY) | 1 | 12 | 0.083 |
| | Estero Tercero (TRC) | 0 | 3 | 0 |
| Delta Edge | Estero Segundo (SGU) | 1 | 2 | 0.500 |
| | Estero Primero (PRI) | 3 | 7 | 0.429 |

spine phenotype (Fig. S1A) is observed in all specimens possessing any of the three *C. hubbsi* RAG1 alleles ($H_1$, $H_2$, or $H_3$).

## Bayesian analyses of microsatellites show evidence of admixture

Based on the evaluation of $\Delta K$, the STRUCTURE analyses on the six microsatellite loci recovered two genetically distinct clusters corresponding to the two species of *Colpichthys* (Fig. 3; for $K = 3$ and $K = 4$, see Fig. S2). Under the admixture model, the $q$-value represents admixture proportion of each individual. We used a $q$-value threshold to distinguish putative purebred and hybrid individuals; for optimal efficiency, this threshold is set to 0.1 (*Vähä & Primmer, 2006*). Populations in the Delta proper—ELZ, SFG, and CHY—were identified as purebred *C. hubbsi*. Several populations away from the Delta such as ANI, NWC, and YAV were identified as purebred *C. regis*. All individuals within or near the western Delta Edge—PRI, SGU, and TRC—show substantial degrees of genetic admixture, having $q$-values between 0.1 and 0.9. Moreover, among these populations there appears to be a gradient in $q$-values as a function of geographic distance from the mouth of the Colorado River. TRC has an average $q$-value of 0.17 while PRI, which is farther from the river mouth, has an average $q$-value of 0.72 (Fig. 3A). A relatively weak signal for admixture in the ADR population beyond the eastern edge of the delta was detected exclusively in the correlated allele frequency model; this signal is absent in the analysis with the independent allele frequency model (Fig. 3B).

## Consilience of phenotypic and nuclear data

Results from separate DAPC analyses were pooled to compare phenotypic and microsatellite variation. Two clusters are supported, consequently only one discriminant function was generated for each of the DAPC analyses. These results show two well separated species with

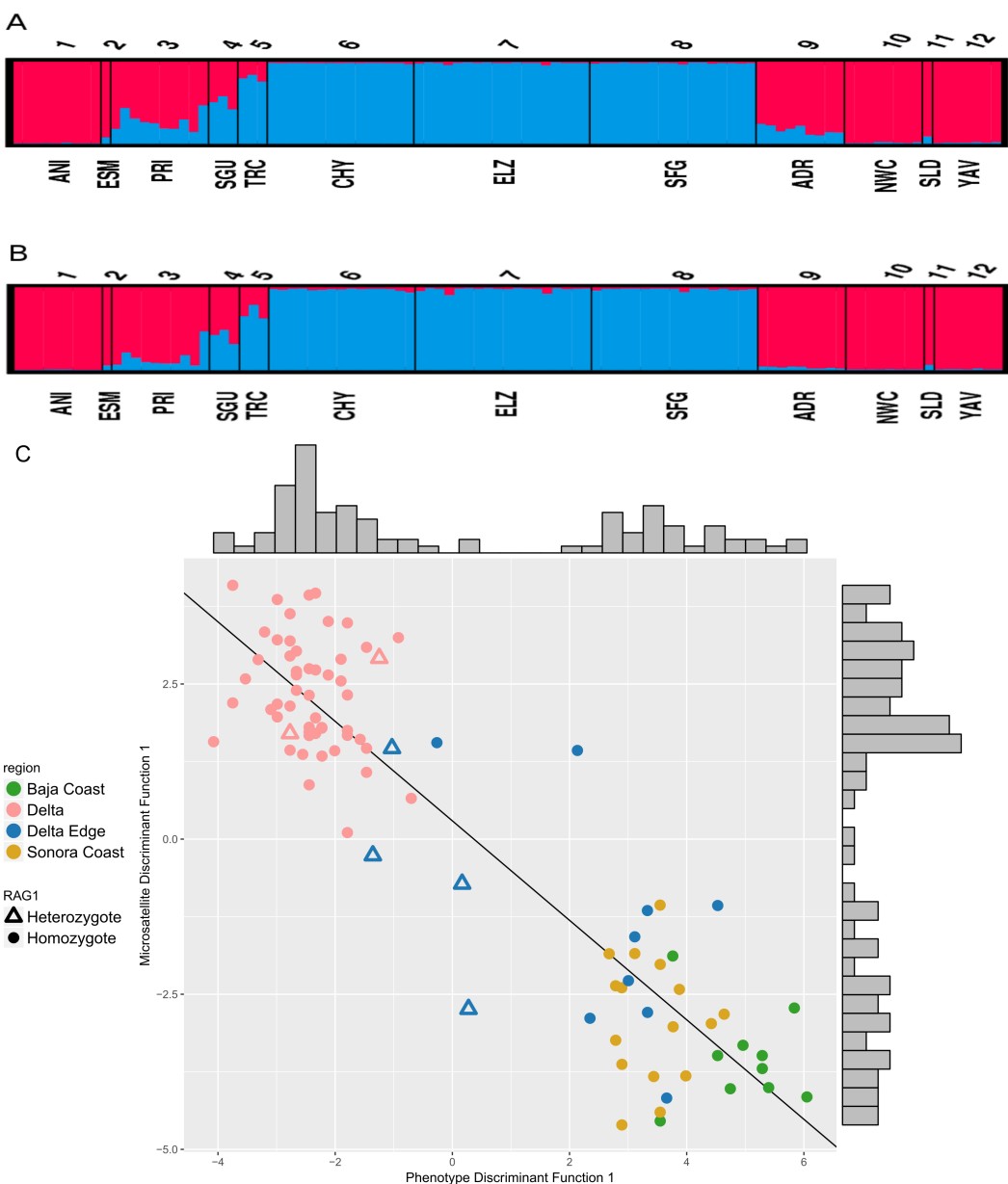

**Figure 3   STRUCTURE assignment of individuals to groups.** STRUCTURE bar plots showing *q*-values of each individual in each sampling locale for *K* = 2. Color for each bar represents admixture proportions of *C. regis* (red) and *C. hubbsi* (blue). Correlated allele frequency model (A). Independent allele frequency model (B). Scatter plot of the first discriminant functions in the DAPC analyses based on phenotypic and microsatellite data (C). Histograms on the top and the right side show the distributions for each axis.

congruent cluster assignment from both data sets (Fig. 3C). Delta Edge specimens preferentially situate between the species clusters. Of the six RAG1 heterozygotes (triangles on Fig. 3C): two recovered in the Delta Edge area fall near the midpoint in morphology between the two species; two recovered in the Delta proper are near the centroid of *C. hubbsi* and the last two from the Delta Edge are phenotypically more similar to *C. hubbsi* than *C. regis*.

## Genetic diversity and demographic history

The Cyt *b* median-joining haplotype networks of the two species are displayed in Figs. 4A and 4B. Twenty-Eight *C. hubbsi* haplotypes are recovered from the four Delta populations; in contrast, only ten haplotypes are found in *C. regis* over a larger, more disjunct geographic range. *C. regis* haplotype diversity differs substantially between the Baja and the Sonoran Coasts; with the exception of one individual from PRI, all non-major haplotypes are found exclusively in Sonora Coast populations. The major *C. regis* haplotype is present in all locations outside the Delta except the two southernmost Sonora Coast sites, SLD and YAV. All five individuals within the Delta proper with the *C. regis* Cyt b sequence possess the major haplotype. Neutrality tests were conducted with the exclusion of the mitochondrial-introgressed individuals; all neutrality tests report significant departure from neutrality. Significantly negative values for Tajima's *D* and Fu's Fs in both species are consistent with either population growth or selective sweep (Table 5). The sum of square deviations and raggedness indices for the mismatch distributions are close to zero and not significant (Table 6), thus we are unable to reject the sudden expansion model for both species. Figures 4C–4D shows the nucleotide mismatch distributions for both species; *C. hubbsi* has a bimodal distribution, while *C. regis* has a unimodal distribution that matches closely with the expected curve under sudden expansion.

## DISCUSSION

### The hybrid zone

*Crabtree (1989)* recognized that the two *Colpichthys* species co-occur in the region just north of San Felipe, designated the Delta Edge in this study. While external morphology of specimens collected in 1968 confirms the co-occurrence of the two species at this region (Table S1), there is no indication from these samples of historical hybridization. Our study presents the first evidence for the modern hybridization of the sister species. STRUCTURE and DAPC analyses strongly support our hypothesis that the Delta Edge region currently represents a hybrid zone. STRUCTURE results limited to the correlated allele frequency model raise the possibility that hybridization may also be occurring on the eastern border of the Delta at Bahía Adair (ADR). The fact that weak admixture signals at ADR are not detected using our other approaches suggests that hybridization at this region, if present, is likely infrequent.

### Directional introgression

In addition to hybridization at the Delta Edge, we detected *C. regis* mitochondrial and RAG1 haplotypes near the mouth and upstream along the Colorado Estuary within the Delta. This could be explained by either the dispersal of first generation hybrids from the hybrid zone to the Delta or by multigenerational backcrossing of hybrids with parental *C. hubbsi* in this case far from the Delta Edge (i.e., introgressive hybridization). Our data lend support to the introgressive hybridization interpretation. Unlike the RAG1 hybrids at the Delta Edge, specimens that possess *C. regis* mitochondrial and/or RAG1 haplotypes within the Delta do not exhibit intermediate morphology. Morphological and microsatellite data for these specimens are typical for *C. hubbsi*. Thus, they likely represent the product

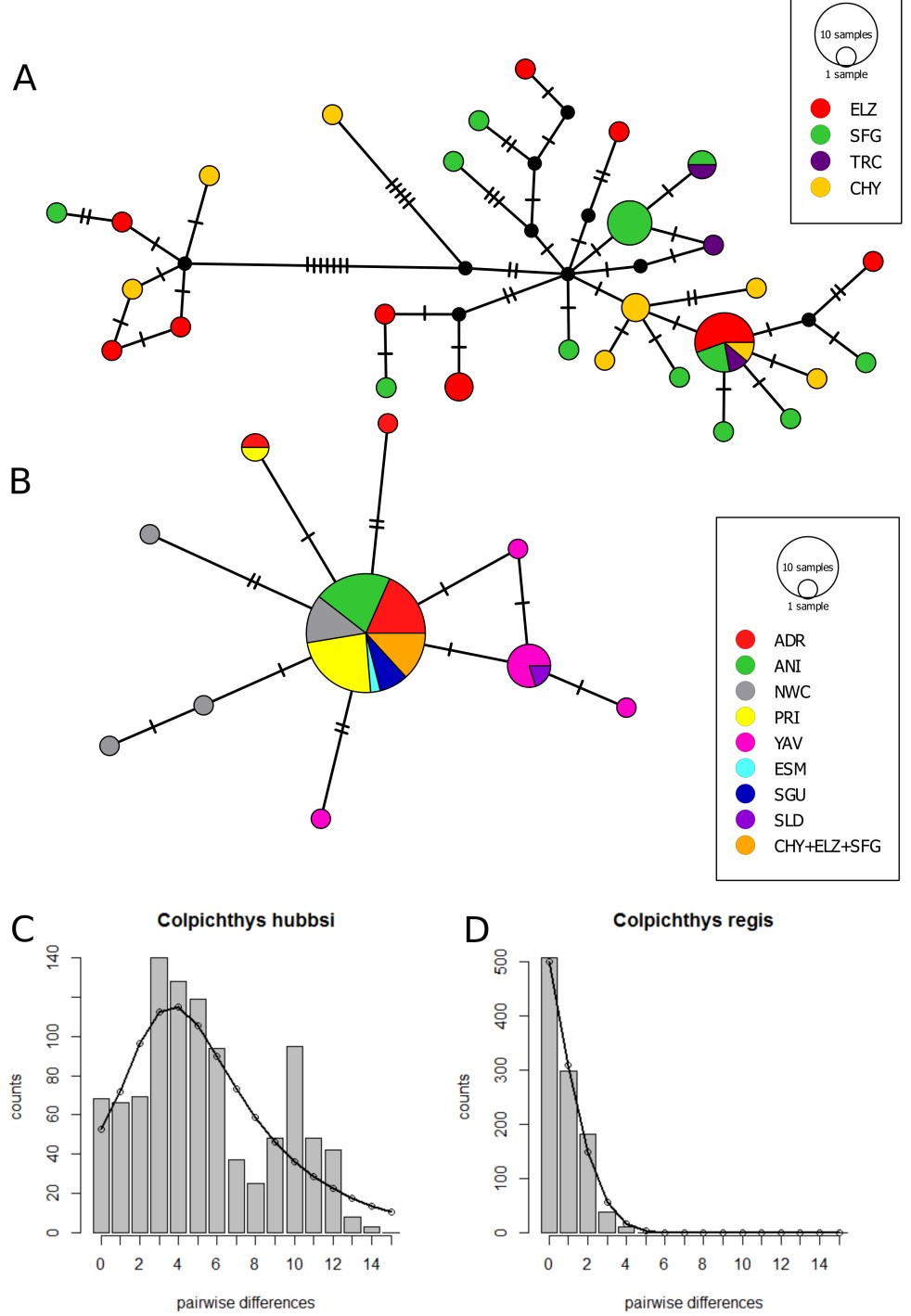

**Figure 4  Cytochrome b haplotype networks.** Cyt b median-joining haplotype networks of *C. hubbsi* (A) and *C. regis* (B). Each circle represents a unique haplotype; size of circles indicates number of specimens with that haplotype. Each tick mark represents a nucleotide difference; black circles represent inferred missing haplotypes. Mismatch distributions for *C. hubbsi* (C) and *C. regis* (D). Histograms represent the observed pairwise differences; black curves represent the expected pairwise differences under the sudden expansion model.

**Table 5  Neutrality tests of Cytochrome b.** Neutrality test results.

| | | | Neutrality tests | | | | |
|---|---|---|---|---|---|---|---|
| | $N$ | $S$ | Tajima's $D$ | Fu & Li's $D^*$ | Fu & Li's $F^*$ | Fu's $Fs$ | $R_2$ |
| *C. regis* | 47 | 11 | **−2.1410**[**] | **−3.3113**[*] | **−3.4502**[**] | **−7.1724**[*] | 0.0514 |
| *C. hubbsi* | 44 | 40 | −1.5645 | −2.2374 | **−2.3783**[*] | **−16.1789**[**] | 0.0596 |

**Notes.**

$N$, number of samples; $S$, number of segregating sites.
Bold: $p < 0.05$.
[*]$p \leq 0.02$.
[**]$p \leq 0.005$.

**Table 6  Mismatch analysis of Cytochrome b.** Mismatch analysis results.

| | | Mismatch analysis (sudden expansion) | | | |
|---|---|---|---|---|---|
| | SSD | Raggedness | $\theta_0$ (99% CI) | $\theta_1$ (99% CI) | $\tau$ (99% CI) |
| *C. regis* | 0.0015 | 0.0779 | 0.018 (0–0.965) | 66,842 (0–99,999) | 1.110 (0–4.605) |
| *C. hubbsi* | 0.0092 | 0.0158 | 1.332 (0–13.558) | 17,339 (6–99,999) | 5.827 (0.105–27.619) |

**Notes.**

SSD, sum of square deviation; CI, confidence interval.

of multiple backcrosses. Thus *C. regis* haplotypes have been introduced into the center of the *C. hubbsi* range via introgressive hybridization. Conversely, we failed to detect any *C. hubbsi* haplotypes outside of the Delta and Delta Edge regions. Thus, introgression does not appear to be reciprocal, although more sampling south of the Delta Edge would allow a stronger conclusion that *C. hubbsi* haplotypes are absent in predominantly *C. regis* populations.

## Parental sex asymmetry in introgression

Phenotypic and microsatellite data (Fig. 3C, Fig. S3) reveal a narrow geographic cline across the 10 km wide hybrid zone which spans three estuaries at the southwestern Delta border—Estero Primero (PRI), Estero Segundo (SGU), and Estero Tercero (TRC). Yet despite morphological and nuclear evidence of hybridization all thirteen specimens from PRI and SGU possess the *C. regis* mitochondrial haplotype. Given matrilineal mitochondrial inheritance, this pattern may indicate hybridization between female parental *C. regis* and male parental *C. hubbsi*. Little is known about the life history and spawning behavior of *Colpichthys*. However, *Crabtree (1989)* noted that the ovipositor of *C. hubbsi* is twice the size of that in *C. regis*, suggesting that differences in reproductive organs could contribute to the directionality of introgression. Behavioral or ecological differences may also play a role, but these hypotheses have yet to be assessed.

## Timing of divergence

Due to the lack of available genetic data from historical samples, the onset of hybridization cannot be easily constrained. However, the ~6% mitochondrial sequence divergence between *C. regis* and *C. hubbsi* suggests that the two species have long been separated with hybridization a relatively recent phenomenon. This mitochondrial divergence is comparable to that of another fish sister species pair, *Gillichthys mirabilis* and *Gillichthys*

*detrusus* (*Swift et al., 2011*), which share distributional patterns with *C. regis* and *C. hubbsi*. Like *C. hubbsi*, *G. detrusus* is narrowly restricted to the silty tidal channels of the Colorado River Delta; its congener, *G. mirabilis*, can be found in the estuaries and lagoons on both coasts of the Gulf. *Swift et al. (2011)* estimated a ∼5 Ma divergence time between *G. detrusus* and *G. mirabilis*. Given that both *C. hubbsi* and *G. detrusus* are Delta endemics, their divergences from their sister species may have been linked to the formation of the Delta estuaries, which occurred after the opening of the Gulf of California. Fossils of *C. regis* have been discovered in the Bouse Formation at the location of Cibola Lake, Arizona (*Todd, 1976*). The Bouse Formation records the changing depositional environments associated with Late Miocene/Early Pliocene progression of the Colorado River to the Gulf (*Poulson & John, 2003*; *McDougall & Martínez, 2014*; *Spencer & Patchett, 1997*; *Roskowski et al., 2010*). Stratigraphic and tectonic reconstructions suggest that the Colorado River reached the Gulf of California by ∼4–5 Ma (*Winker & Kidwell, 1986*; *Dorsey et al., 2007*; *Spencer et al., 2013*; *Crossey et al., 2015*; *Howard et al., 2015*). Based on this chronology, speciation leading to the Delta endemics presumably followed the establishment of the Delta habitats around this time (*Swift et al., 2011*; *Ellingson, 2012*).

## Selection and ecological speciation in response to salinity conditions

The parapatric distribution of *Colpichthys* in the Northern Gulf suggests that the two species may have diverged due to ecological differentiation reinforced by divergent selection on local adaptations. The divergence is likely linked to the gradient in salinity in the Northern Gulf. While virtually no physiological data exists for *C. hubbsi*, *C. regis* is known to tolerate hypersaline conditions as it is frequently observed in lagoons with salinity values reaching 50 psu (*Castro-Aguirre & Espinosa Pérez, 2006*). Prior to the completion of Hoover Dam in 1935, the water output at the Delta was estimated to be 16 to 18 billion cubic meters per year (*Stockton & Jacoby Jr, 1976*), supporting a substantial historic fresh to salt water gradient through much of the Delta. Development of an inverse hypersaline estuary followed in the 20th Century damming and water removal from the system (*Lavín & Sánchez, 1999*). Although we are not aware of any studies of salinity tolerance in captivity for *Colpichthys*, as noted above, silversides have developed ecological species in response to salinity gradients in Brazil and Australia, the Mediterranean and the Eastern US (*Bamber & Henderson, 1988*; *Beheregaray & Sunnucks, 2001*; *Fluker, Pezold & Minton, 2011*; *Johnson, Watts & Black, 1994*; *Klossa-Kilia et al., 2007*; *Francisco et al., 2006*; *Olsen, Anderson & McDonald, 2016*; *Trabelsi et al., 2002*). Thus, it is likely that historic salinity gradients in the Delta played a role in reproductive isolation between the *Colpichthys* sister species. Moreover, selection maintaining these species is prone to breakdown following the loss of freshwater.

Due to the difference in density of fresh and saltwater, salinity is known to influence the morphology of the gas bladder, which maintains neutral buoyancy in teleost fishes. Among its closest relatives in Atherinopsinae, *C. hubbsi* is the only species where the gas bladder does not extend posterior to the visceral cavity (*Crabtree, 1989*), suggesting that salinity related selection impacts the species. Other potential selective factors may involve the unusual silty conditions and tides in the Delta and the limited depth in the Delta. Relative to other examples of species boundaries particularly maintained by selection and
differences in salinity in fishes such as those associated with the transition into the Baltic, which developed during the Holocene (*Barrio et al., 2016*); or those in Brazilian coastal silversides where divergence of freshwater forms was in the Pleistocene (*Beheregaray, Sunnucks & Briscoe, 2002*), our inferred early Pliocene timing of speciation based on sequence differences between *C. regis* and *C. hubbsi* is substantially more ancient.

## Evolutionary implications of introgression for colpichthys

Introgressive hybridization potentially has a number of distinct causes, including anthropogenic introduction and habitat modification. It also has a number of different evolutionary outcomes ranging from sharing of advantageous alleles between species (*Martin, Bouck & Arnold, 2006*; *Whitney, Randell & Rieseberg, 2010*; *Pardo-Diaz et al., 2012*; *Hedrick, 2013*) to extinction (*Rhymer & Simberloff, 1996*; *Todesco et al., 2016*). Of particular interest here are examples of habitat alteration that lead to "speciation reversal", or the introgressive loss of ecological speciation. Eutrophication, for example, has led to numerous examples of species loss in fishes (*Vonlanthen et al., 2012*; *Seehausen, Van Alphen & Witte, 1997*). Given that salinity is associated with ecological species in a number of silversides, and the loss of freshwater in the Colorado Delta, introgression into the more freshwater adapted *C. hubbsi* from the marine adapted *C. regis* would be expected. Ultimately, introgression of *C. regis* alleles adapted for higher salinity could leave *C. hubbsi* at risk of genetic swamping. In the long run, the risk of extinction by speciation reversal for *C. hubbsi* will depend on the extent to which the loss of salinity gradient has eliminated the habitat distinction in the Northern Gulf. While this study only identified backcrossed hybrids based on the limited genetic loci sampled, it is possible that many of the morphological "pure bred" *C. hubbsi* are in fact "cryptic" hybrids. Depending on the frequency and relative abundance of such "cryptic" hybrids, interspecific gene flow may be more pervasive than suggested by our data. To assess whether "speciation reversal" could pose a threat to *C. hubbsi*, future whole-genome analyses will be required to quantify the amount of genomic introgression as well as identify loci that may remain under divergent selection in the face of gene flow.

## Demographic expansions

Given the geographic restriction of *C. hubbsi*, one immediate concern is whether the reduced Colorado River outflow has directly contributed to a population decline. The rise in salinity at the Delta has been implicated in the local decline of the clams in the genus *Mulinia* (*Rodriguez, Flessa & Dettman, 2001*). Based on our results from neutrality tests and mismatch analyses, we find significant departure from neutrality, but no evidence to support a recent bottleneck in *C. hubbsi*. On the other hand, contrary to initial expectation, *C. regis* exhibits surprisingly low genetic diversity compared to its geographically restricted congener. This pattern is seen in all three types of molecular markers (Cyt *b*: Fig. 4; RAG1: Table 3; microsatellites: Table S2). The results from this study are consistent with demographic expansions in the recent past. Other factors, such as selective sweep or fine-scaled population structure (*Ptak & Przeworski, 2002*), could produce similar mtDNA patterns, but these mechanisms seem unlikely to have generated the similar patterns of

diversity in all three data sets. We offer the following scenario as a possible explanation for the observed data. In the mismatch analysis, the model parameter $\tau$ estimates the age of demographic expansion. Based on this, our results (Table 6) suggest that *C. hubbsi* may have attained larger populations in the Delta estuarine habitats than *C. regis* and that *C. hubbsi* populations were either more continuous or experienced a more ancient expansion relative to *C. regis*. One possible explanation for this could come from Pleistocene climate and sea-level fluctuations. Recent work (*Dolby et al., 2016*) demonstrated that interactions between coastal geomorphology and Pleistocene sea-level fluctuations limited estuary habitats to refugia during glacial sea-level minima, impacting modern population structures of estuarine fish. In addition, during the Pleistocene, wetter climates increased Colorado River outflow into the Gulf, while the gentle sloping bathymetry of the Northern Gulf together with the lowering of sea-level likely expanded habitats for *C. hubbsi* much further south, allowing for population growth. Meanwhile, the steeper outer shelves on both sides Gulf to the south could have restricted suitable refugial habitats for *C. regis* during sea-level lowstand (*Dolby, 2015*). The lack of genetic diversity in *C. regis* populations may represent rapid range expansion from these limited refugia following sea-level rise at the end of the Pleistocene. Crucial to this hypothesis is the precise timing of the demographic and range expansions. More complete sampling of *C. regis* within each of the modern localities or genomic approaches are needed in future studies to further explore this issue.

## Potential implications for Delta endemics

Of the numerous Gulf endemics (*Palacios-Salgado et al., 2012*), some are confined to the northernmost Gulf, and a smaller subset are very locally endemic to the Delta itself, which is an area protected as a Biosphere Reserve (Reservade la Biosfera del Alto Golfo de California y Delta del Rio Colorado). Of these northernmost endemics, some are relicts of broader distributions, while others appear to be ecological species that evolved in association with the Colorado Delta (*Jacobs, Haney & Louie, 2004*). Apparent relictual species include the critically endangered Vaquita, the world's smallest cetacean, whose closest relative is found on the coasts of South America south of Peru (*Munguia-Vega et al., 2007*). The totoaba (*Totoaba macdonaldi*) was historically an important fishery resource that appears to have depended on freshwater input from the Colorado River for rapid growth early in development (*Rowell et al., 2008*). This genus, broadly endemic to the Gulf, is known from fossils in the Miocene of California Central Valley (*Huddleston & Takeuchi, 2007*); its closest relatives are in the Atlantic and IndoPacific (*Lo et al., 2015*). In addition to these relicts which may have been directly impacted by salinity changes or other anthropogenic activities such as overfishing, three taxa in the region appear to have evolved ecological species pairs associated with the Delta. Apart from the fish genera *Colpichthys* and *Gillichthys* as discussed above, the fiddler Crab *Uca monilifera* and *U. princeps* conform to the same pattern (*Brusca, 1980*). Thus, out of a suite of five northernmost Gulf endemic estuarine species, three are potentially at risk of introgression due to hydrologic modification. While a number of studies have argued that the northern Gulf fauna has been directly impacted by salinity (*Rodriguez, Flessa & Dettman, 2001*; *Rowell et al., 2008*), others have emphasized the minimal change in Northern Gulf biological productivity associated with

the anthropogenic loss of freshwater input from the Colorado River (*Brusca et al., 2017*). However, there may nevertheless be strong impacts on the endemic diversity of this biologically unique region, especially with regards to introgression in endemics specific to the Colorado Delta proper.

## Global change context

Early 20th century water removal tended to impact larger river systems primarily in semi-arid settings, where freshwater is at a premium and where economic development is well advanced; this is well exemplified by the Southwestern US. With increasing population growth and economic development in a wider range of geographic settings, damming and diversion efforts have continued to expand from rivers that traverse arid regions, impacting the Deltas estuaries of rivers such as the Nile (*Stanley & Warne, 1998*) and the Euphrates (*Isaev & Mikhailova, 2009*), to many of the largest river systems of Asia, such as the Mekong (*Kondolf, Rubin & Minear, 2014*). Despite this, the impact on estuarine diversity, especially as regards to the potential loss of diversity due to introgression between ecological species, has not been widely discussed or assessed. In the case of the Colorado River Delta, we have been able to establish that introgression is ongoing in the context of the ecologically altered Delta environment. This is the first step in establishing the risk of introgressive species loss in river estuaries that have suffered from water extraction and loss of freshwater input. This is yet another factor in a long list of ecological and biodiversity impacts consequent to altering riverine hydrologic processes, and should be systematically considered in a global context so that the full ecological consequences and diversity costs of such development can be properly understood.

## CONCLUSION

We document the Pliocene separation and recent introgression between a sister species pair in the northern Gulf of California. Our evidence supports hybridization and introgression into the IUCN endangered Delta endemic *Colpichthys hubbsi* from the more widespread Gulf species *C. regis* following 20th century environmental alteration through water removal and salinity change in the Delta system. The work combines multiple lines of phenotypic and genetic evidence to demonstrate this directional introgressive hybridization. We find no evidence for population decline in *C. hubbsi*, yet continued introgression may jeopardize the genetic integrity of this taxon. Genetic diversity of *C. regis* is revealed to be lower than expected for a healthy, widely distributed species so it likely merits its IUCN threatened designation. Our work highlights the need for closer monitoring and further demographic investigations of *Colpichthys* populations not just within the Delta but also throughout the Northern Gulf. More generally, our work implies that other Colorado Delta-specific endemics such as the goby *Gillichthys detrusus* and the fiddler crab *Uca monilifera* may also be ecological species at risk of loss through introgression from their proximally distributed congeners. More broadly, a general suite of ecological species around the world that evolved in response to salinity gradients in major river systems are likely at increasing risk of introgressive extinction as water extraction and climate change impacts accelerate in river systems globally.

## ACKNOWLEDGEMENTS

The authors would like to thank Kirk Lohmueller, Blaire Van Valkenburgh, and Jonathan Richmond for feedback and comments; Rick Feeney of the Los Angeles County Natural History Museum for access to an X-ray machine and the specimens in the ichthyology collection; HJ Walker and Phil Hastings of the Scripps Institution of Oceanography for providing the historical specimens; Greer Dolby and Ryan Ellingson for advice on data analysis; Bruno Passerelli for assistance in data collection; Lloyd Findley for facilitating permitting and collection. Assistance with field sampling was provided by T Baumiller, V Cassano, R Ellingson, R Hechinger, F Hertel, J Lorda, and D Yuan.

### Funding

The authors received no funding for this work.

### Competing Interests

The authors declare there are no competing interests.

### Author Contributions

- Clive L.F. Lau conceived and designed the experiments, performed the experiments, analyzed the data, wrote the paper, prepared figures and/or tables, reviewed drafts of the paper.
- David K. Jacobs conceived and designed the experiments, contributed reagents/materials/analysis tools, wrote the paper, prepared figures and/or tables, reviewed drafts of the paper.

### Field Study Permissions

The following information was supplied relating to field study approvals (i.e., approving body and any reference numbers):

Collections were carried out under Mexican federal collecting permit (Permiso de Pesca de Fomento) DGOPA 14253. 101005.6950, and its extension DGOPA 06435.210606.2640, issued to Findley and Jacobs by the Comisión Nacional de Acuacultura y Pesca of the Secretaría de Agricultura, Gana- dería, Desarrollo Rural, Pesca y Alimentación (SAGARPA).

### DNA Deposition

The following information was supplied regarding the deposition of DNA sequences:

Data are included as a Supplemental File.

### Data Availability

Github: https://github.com/clflau/Colpichthys_github/.

### Supplemental Information

Supplemental information for this article can be found online at http://dx.doi.org/10.7717/peerj.4056#supplemental-information.

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
