# Peer review of "Introgression between ecologically distinct species following increased salinity in the Colorado Delta- Worldwide implications for impacted estuary diversity"

_PeerJ, doi:10.7717/peerj.4056_

## Round 0.1 · original submission · Minor Revisions

· Academic Editor

Minor Revisions

All three reviewers recommended that your paper be accepted, and two had specific recommendations for changes or clarifications. These should be easy to incorporate in a final copy.

·

Basic reporting

No comment

Experimental design

No comment

Validity of the findings

No comment

Additional comments

In this paper, Lau and Jacobs report morphological and genetic data that clearly separate the northern Gulf of California sister species, Colpichthys regis and C. hubbsi, and provide evidence for introgressive hybridization between the two distinct species in the Delta region. The paper was very well written and easy to follow. Analyses and presentation of both morphological and genetic data were appropriate and quite thorough, and supported the authors' main conclusions. I only have a couple of comments that I feel the authors might want to address. Although seemingly trivial, the name "Scripps Institute of Oceanography" is not correct; "Institute" should be changed "Institution" in both the text and supplementary material.
The second comment regards the bimodal distribution of the mismatch distribution of C. hubbsi (Fig. 4C). Although the SSD and raggedness statistics were not significant, and the sudden expansion model could not be rejected for either species, these metrics are known to have low statistical power. The mismatch plot of C. hubbsi would suggest a stable population. It might be worthwhile emphasizing this a bit more (e.g. line 313), especially since the side-by-side plots of C. hubbsi and C. regis clearly show two distinct patterns, which are consistent with the two distinct haplotype networks.

·

Basic reporting

One relevant reference could be cited: Hastings, P. A. & L. T. Findley. 2006. Marine Fishes of the Biosphere Reserve, Northern Gulf of California. Pp. 364-382. In: Felger, R. & W. Broyles (eds). Dryborders: Great Natural Areas of the Gran Desierto and Upper Gulf of California. Univ. Utah Press, Salt Lake City, Utah.

Otherwise, no comment

Experimental design

no comment

Validity of the findings

no comment

Additional comments

This manuscript is well-written and data are thoroughly analyzed and conclusions are for the most part fully justified. It represents an important contribution with relevance not only the system under study but also to similar estuarine systems around the world.
The attached pdf has a number of questions, mostly of clarification, suggested rewording, or request for supporting references.

Reviewer 3 ·

Basic reporting

Meets all standards.

Experimental design

Meets all standards.

Validity of the findings

Meets all standards. Impact and novelty is well assessed.

Additional comments

This is a well conceived, written, and executed paper on an extremely important topic (the loss of estuarine habitat due to human water usage). It should be published as is (except a minor misspelling of Colpichthys in the abstract)

---

## Round 0.2 · accepted · Accept

· Academic Editor

Accept

Nice paper! I think it will be widely read and cited.